# The Role of Drug Resistance in Candida Inflammation and Fitness

**DOI:** 10.3390/microorganisms13081777

**Published:** 2025-07-30

**Authors:** Gabriella Piatti, Alberto Vitale, Anna Maria Schito, Susanna Penco, Daniele Saverino

**Affiliations:** 1Department of Surgical Sciences and Integrated Diagnostics (DISC), University of Genoa, 16126 Genoa, Italy; anna.maria.schito@unige.it; 2Unit of Microbiology, IRCCS Ospedale Policlinico San Martino, 16132 Genoa, Italy; alberto.vitale@hsanmartino.it; 3Department of Experimental Medicine (DiMeS), University of Genoa, 16126 Genoa, Italy; susanna.penco@unige.it (S.P.); daniele.saverino@unige.it (D.S.)

**Keywords:** *Candida*, *C. auris*, fungal fitness, drug resistance, fluconazole, innate immunity, cytokine

## Abstract

Drug resistance in *Candida* may result in either a fitness cost or a fitness advantage. *Candida* auris, whose intrinsic drug resistance remains unclear, has emerged as a significant human pathogen. We aimed to investigate whether *Candida* fitness, including early interaction with the host innate immune system, depends on the antifungal susceptibility phenotype and *putative-*associated resistance mutations. We compared interleukin-1β, interleukin-6, interleukin-8, and tumor necrosis factor α production by human colorectal adenocarcinoma cells stimulated by fluconazole-susceptible and fluconazole-resistant strains of *Candida albicans*, *C. parapsilosis*, *C. tropicalis*, and *C. glabrata*, as well as fluconazole-resistant *C. auris* strains. Sensitive *Candida* strains induced lower cytokine levels compared with *C. auris* and resistant strains, except for TNF a. Resistant strains induced cytokine levels like *C. auris*, except for higher IL-1β and lower TNF-α. Susceptible strains exhibited cytokine profiles distinct from those of resistant strains. *C. auris* induced cytokine levels comparable to resistant strains but displayed profiles resembling those of susceptible strains. This study highlights the relationship among antifungal susceptibility, fungal fitness and host early immunity. *C. auris* behavior appears to be between fluconazole-sensitive and fluconazole-resistant strains. Understanding these dynamics may enhance the knowledge of the survival and reproduction of resistant Candida and the epidemiology of fungal infections.

## 1. Introduction

Most fungal infections worldwide are superficial diseases of skin and mucosa in humans and affect patients in the community [1]. The most invasive and serious infections caused by yeasts, such as sepsis, occur in hospital settings and are caused by *Candida* species. These are commensal organisms and opportunistic pathogens whose invasive infections are strongly linked to the frailty of the host and life-threatening diseases [2].

Epidemiology studies in humans have demonstrated that the numerous virulence factors of *Candida* are effective in causing invasive infections only when host immune surveillance, typically maintained by the structural and immunological integrity of epithelial and mucosal barriers, is compromised [3,4]. Innate immunity is fundamental to maintaining the symbiotic relationship between the host and commensal microorganisms, among which yeasts are an integral component [5]. The protective response begins with the interaction between pattern recognition receptors (PRRs) of host cells and the conserved microbe-associated molecular patterns (MAMPs), leading to the production of inflammatory cytokines aimed at recruiting phagocytes and eliminating microorganisms [6,7]. Mannans, phospholipomannan, and β-glucans of the fungal cell wall have been identified as MAMPs, mediating the early stages of host–pathogen interactions in animals [8]. In addition, ergosterol of the fungal membrane has been identified as a MAMP, mediating the host–pathogen interactions in plants [9,10,11].

The antifungal treatment, which is limited by the evolutionary proximity of fungi to higher eukaryotes and the resulting scarcity of specific drug targets, is further hampered by the increasing phenomenon of resistance [12]. Fluconazole, a member of the azole class, has been, and remains, the most widely used therapeutic option [12]. Azoles inhibit the synthesis of ergosterol, a component specific to the fungal cell membrane, by targeting cytochrome P450-dependent lanosterol 14-α-demethylase and sterol Δ-5,6 desaturase [13]. Having no candidacidal activity, fluconazole is particularly prone to selecting inherently less susceptible species and promoting the emergence of new resistance mechanisms [14]. Several point mutations may be present alone or simultaneously in individual fluconazole resistant *Candida* isolates, resulting in a reduction in the intracellular concentration of fluconazole or a reduction in the affinity of the drug for the molecular target, or even resulting in the overexpression and increased function of enzymes in the metabolic pathway of ergosterol synthesis [15,16,17].

The ability of a microbial population to survive and reproduce is defined as fitness. Although viral, bacterial, and fungal genotypes that are resistant to drugs are expected to have a selective advantage over susceptible genotypes in the presence of antimicrobial agents, resistance may incur fitness costs [14,18,19,20]. The persistence and dissemination of drug-resistant organisms in nature depend on the relative fitness of drug-sensitive and drug-resistant genotypes [21]. The genomic plasticity and adaptation of yeasts, which exceed those of viruses and bacteria, make the problem of resistance and the associated fitness cost more difficult to interpret [14,22]. Several studies have reported divergent findings: both advantages and disadvantages in fungal fitness associated with resistance phenotypes, as measured in various functional experimental assays [13,21,23].

We aimed to investigate the fitness of *Candida* strains by examining potential differences between fluconazole-sensitive and fluconazole-resistant *Candida* strains in their interaction with cells involved in the early immune response, assessing the induction of pro-inflammatory cytokine.

In vitro we stimulated the human colorectal adenocarcinoma epithelial cell line DLD-1 with 91 *Candida* isolates, categorized as fluconazole-susceptible, fluconazole-resistant, and *C. auris* strains. The isolates belonged to *C. albicans*, *C. parapsilosis*, *C. tropicalis*, and *C. glabrata* species, in which azole resistance is known to develop through stepwise selection, and to the *C. auris* species, which is considered intrinsically multidrug-resistant [12]. Following the stimulation, we assessed pro-inflammatory cytokine production, i.e., the levels and profiles of interleukin-1β (IL-1β), interleukin-6 (IL-6), interleukin-8 (IL-8), and tumor necrosis factor-α (TNF-α).

## 2. Material and Methods

### 2.1. Candida Species Isolates

*Candida* isolates came from the collection of clinical strains, all kept frozen at −20 °C in 20% glycerol Luria broth (Sigma-Aldrich). We enrolled ninety-two *Candida* strains including eighteen *C. albicans*, thirty-nine *C. parapsilosis*, nineteen *C. tropicalis*, four *C. glabrata*, and twelve *C. auris* strains. All the strains were isolated from blood cultures.

### 2.2. Measure of Drug Susceptibility Phenotype of Candida Strains

The phenotype of susceptibility towards antifungal drugs was measured with a commercial microdilution assay (Sensitre YeastOne ITAMYUCC, by Thermo Fisher Scientific, Monza, Italy) following the manufacturer’s instructions and, with the exception of *C. auris*, compared with the clinical break point updated in 2022 by the CLSI reference study group [24]. The test evaluates the minimal inhibiting concentration (MIC) of polyenes (amphotericin B), azoles (isavuconazole, posaconazole, fluconazole, voriconazole, itraconazole), and echinocandins (micafungin, caspofungin, anidulafungin).

### 2.3. Cell Culture

DLD-1-CCL-221 (a colorectal adenocarcinoma cell line isolated from the large intestine of a colon adenocarcinoma patient expressing TLR-1, TLR-2, TLR-4, TLR-5, and TLR-8, in addition to other innate immune receptors, such as RIG-I-like, NOD-like, and C-type lectin receptors) was utilized for stimulation experiments and purchased from ATCC (https://www.atcc.org/, accessed on 5 May 2025). Cells were cultured in RPMI medium (Gibco) supplemented with 10% heat-inactivated fetal bovine serum (Sigma), free of antibiotics, at 37 °C in a humid atmosphere of 95% air and 5% CO_2_. Forty-eight hours before fungal stimulation, DLD-1 cells (1 mL) were seeded into 24-well plates at 1.5 × 10^5^ per well and allowed to grow to confluency over two days. On the day of the assay, RPMI on confluent cell monolayers was removed by suction, cells were washed with sterile phosphate-buffered saline (PBS), and 1 mL of fresh RPMI complete medium, as subsequently described, was applied.

### 2.4. DLD-1 Stimulation with Candida Strains

All clinical *Candida* strains, grown overnight in 5 mL Luria broth (Sigma-Aldrich, Milano, Italy), were assessed using a Vitek Densichek nephelometer (Biomerieux, Firenze, Italy) and diluted to a concentration of 0.5 McF (10^8^ CFU mL^−1^) with PBS [25]. Then, 10 mL of each fungal suspension was added to each well containing DLD-1 cells to achieve a final concentration of 10^6^ CFU mL^−1^. Uninfected cells were used as the baseline for the cytokine production assay. The fungi were spun onto the cells by a 3 min 400× *g* centrifugation and incubated for 6 and 24 h at 37 °C. The fungal concentration and the incubation time were chosen to minimize cytotoxicity to the colorectal adenocarcinoma cell line, which was checked by trypan blue staining, and to obtain an optimal fungal stimulation with respect to baseline cytokines and cytokine production. Briefly, cell death involved less than 2% of the cells at 6 h from the start of stimulation and less than 15% after 24 h. No significant differences in cell death were observed across experimental conditions, preventing a direct correlation between cytokine production and cell viability (Figure 1). After incubation, the supernatants were collected, centrifuged to remove fungi and cell debris, and stored at −80 °C until they were assayed using a human specific ELISA determination method.

### 2.5. Measurement of Cytokines Following DLD-1 Stimulation with Candida Strains

To quantify levels of IL-1β, IL-6, IL-8, and TNF-α after 6 and 24 h stimulation, culture supernatants obtained from duplicate experiments were analyzed by specific sandwich enzyme-linked immunosorbent assays (ELISAs). Once the collection of supernatants was cleared by centrifugation, cytokines were measured in triplicate using an ELISA kit (ImmunoTools GmbH, Friesoythe, Germany; IL-1β, catalogue number: 31670019; IL-6: 31670069U1; IL-8: 31670089U1; TNF-α: 31673019U1) according to the manufacturer’s instructions. The minimum detectable concentrations were 18 pg/mL for IL-1β, 6.1 pg/mL for IL-6, 2.6 pg/mL for IL-8, and 22 pg/mL for TNF-α, as reported by the manufacturer. The experiments were repeated three times, and the means ± standard deviations (SDs) were presented. For all the ELISA kits, the intra-assay precision was <8%, and the inter-assay precision was <10%.

### 2.6. Statistical Analyses

All the experiments in this study were performed three times. Experimental values were given as means ± SD. For statistical analysis, we considered the cumulative amount of each cytokine induced by a single species. The statistical significance of differences between the control values and the test values was evaluated using a one-way ANOVA. Normality assumptions were verified using the Shapiro–Wilk test. All the assumptions were fulfilled. For analyses of correlation between different cytokine levels referred to different *Candida* strains, we performed a Spearman test. *p*-values were declared significant at 0.05. Data were analyzed using the GraphPad Software version 10.5 (Boston, MA, USA).

## 3. Results

### 3.1. Candida Species, Drug Susceptibility Phenotypes, and Definitions

Table 1 shows the MICs of fluconazole and amphotericin B for ninety-two *Candida* strains (*C. albicans*, *C. parapsilosis*, *C. tropicalis*, *C. glabrata*, *C. auris*), as well as the MICs of those drugs, plus voriconazole and caspofungin for *C. auris* strains. Other drugs, not shown, including echinocandin, had an MIC under the breakpoint or the epidemiological cutoff value (for amphotericin B, whose MICs are not yet defined) for all strains of all species. Sixteen *C. albicans* were susceptible and two were resistant to fluconazole, while all had MIC of amphotericin B ≤ the epidemiological cutoff value. Eighteen *C. parapsilosis* strains were susceptible to fluconazole with an MIC of amphotericin B ≤ the epidemiological cutoff value; twenty-one strains were resistant to fluconazole, among which five had an MIC of amphotericin B > the epidemiological cutoff value (MICs 2 mg/L and 4 mg/L). The nineteen *C. tropicalis* and the four *C. glabrata* strains were susceptible to all antifungals tested. All fluconazole susceptible strains were susceptible to all the azoles. The interpretation of MICs values for *C. auris* is not yet defined [24]. C. auris is an emerging multidrug-resistant yeast, and its classification as resistant or sensitive is based on tentative MIC breakpoints, since no species-specific clinical breakpoints have been formally established by the Clinical and Laboratory Standards Institute [24]. We reported the MICs of fluconazole, amphotericin B, voriconazole, and caspofungin towards *C. auris* to allow comparison with breakpoints x or with epidemiological cut-off values, typically maintained by the structural and immunological integrity of epithelial and mucosal barriers [24]. All *C. auris* strains showed fluconazole MICs exceeding 256 mg/L (see Table 1 for MICs of other antifungals). The susceptible strains to all fungal drugs were classified as sensitive, whereas resistant strains to fluconazole were classified as resistant. Finally, *C. auris* strains were analyzed as a separate group due to their inherently uncertain drug resistance profile.

### 3.2. Quantity of Cytokines Induced over 6 and 24 h by Candida Strains According to the Global Fluconazole Susceptibility Phenotypes

We stimulated DLD-1 cells with *Candida* strains from all species, grouped as ergosterol-susceptible (*C. albicans*, *C. parapsilosis*, *C. tropicalis*, and *C. glabrata*), ergosterol-resistant (*C. albicans* and *C. parapsilosis*), and *C. auris* strains. The induction of IL-1β, IL-6, and IL-8 by susceptible strains was lower than that induced by the resistant strains and by *C. auris* after both stimulation times points. Figure 2 summarizes the production of pro-inflammatory cytokines by DLD-1 cells after 6 hours of induction.

As can be depicted, the population of resistant strains exerts a dual effect on IL-1β production by DLD-1 cells after 6 h of stimulation. Accordingly, resistant *Candida* strains can be subdivided into two groups based on their ability to induce IL-1β production. Strains inducing low IL-1β levels (<100 pg/mL) behave similarly to susceptible strains and *C. auris*. Conversely, strains inducing high IL-1β production (>250 pg/mL) exhibit a distinct response compared with susceptible strains and *C. auris*. However, when considered as a unique group, resistant strains induce IL-1β production at levels that are statistically different from those induced by susceptible strains and by *C. auris*.

Finally, the heterogeneity in inducing IL-1β production suggests an early divergence in the inflammatory response. However, by 24 h, this distinction was no longer evident, as the populations appeared to converge into a single homogeneous group with no visibly discernible differences in IL-1β induction (compare Figure 2 to Figure 3).

IL-1β production induced by susceptible strains (53.93 pg/mL +/− 41.59) was significantly lower than by resistant ones (259.7 pg/mL +/− 170.8, *p*-value < 0.001) and by *C. auris* (102.3 pg/mL +/− 77.19, *p*-value = 0.021). IL-1β levels induced by resistant strains were significantly higher than those induced by *C. auris* (*p*-value = 0.017). The amount of IL-6 induced by susceptible strains (18.90 pg/mL +/− 5.53) was significantly lower than by resistant ones (34.42 pg/mL +/− 8.01, *p*-value < 0.001) and by *C. auris* (37.49 pg/mL +/− 11.45, *p*-value < 0.001). The inductions of IL-6 by resistant and *C. auris* strains were similar (*p*-value = 0.707). A similar trend was observed for IL-8: susceptible strains induced lower levels (17.19 ± 6.34 pg/mL) compared with resistant strains (32.41 ± 5.84 pg/mL, *p* < 0.001) and C. auris (29.56 ± 8.39 pg/mL, *p* < 0.001), while IL-8 levels induced by resistant and C. auris strains were comparable (*p* = 0.338). TNF-α production by resistant strains (160.3 ± 68.86 pg/mL) was similar to that induced by susceptible strains (140.60 ± 63.54 pg/mL, *p* = 0.211), but significantly lower than that induced by C. auris (275.0 ± 110.9 pg/mL, *p* = 0.006). TNF-α levels induced by susceptible strains were also significantly lower than those induced by *C. auris* (*p* < 0.001).

Figure 3 summarizes the production of pro-inflammatory cytokines by DLD-1 cells after 24 h of stimulation, showing an overall increase and similar pattern to that observed after the 6 h.

IL-1β levels induced by susceptible strains (115.5 ± 62.86 pg/mL) remained significantly lower (*p* < 0.001) compared with those induced by resistant strains (534.9 ± 265.3 pg/mL) and *C. auris* strains (363.6 ± 224.1 pg/mL). The higher IL-1β induction by resistant strains compared with *C. auris* persisted after prolonged incubation (*p* = 0.033). IL-6 and IL-8 induction patterns at 24 h were similar to those observed at 6 h. Specifically, IL-6 levels induced by susceptible strains (49.75 ± 15.64 pg/mL) were significantly lower than those induced by resistant strains (177.6 ± 74.61 pg/mL, *p* = 0.003) and *C. auris* (145.9 ± 278.1 pg/mL, *p* < 0.001). IL-6 induction by resistant and *C. auris* strains remained comparable (*p* = 0.190). IL-8 levels induced by susceptible strains (34.66 ± 16.92 pg/mL) were significantly lower than those induced by resistant strains (62.0 ± 19.47 pg/mL, *p* = 0.001) and *C. auris* (54.78 ± 11.77 pg/mL, *p* < 0.001). IL-8 induction by resistant and *C. auris* strains remained similar (*p* = 0.283). After prolonged stimulation, TNF-α levels induced by resistant strains increased (634.8 ± 278.1 pg/mL), becoming comparable to those induced by *C. auris* (598.8 ± 222.0 pg/mL, *p* = 0.615), while remaining statistically similar to those induced by susceptible strains (462.5 ± 139.7 pg/mL, *p* = 0.148). TNF-α levels induced by susceptible strains were significantly lower than those induced by *C. auris* (*p* = 0.006).

### 3.3. Comparison Between Cytokines Levels Induced by Different Candida Species Within Homogenous Susceptibility Phenotype

To ensure that the observed data were not due to random variation, we assessed whether the stimulatory activities of different *Candida* species within each susceptibility phenotype did not differ significantly. This evaluation was necessary due to the reported differences in innate immune recognition of different *Candida* species by human peripheral blood phagocytes, as demonstrated in previous studies [26,27].

Table 2 indicates that forty-six out of fifty-six comparisons yielded consistent results, with *p*-values > 0.05. Specifically, within the group of susceptible phenotypes, the comparisons between different species for IL-1β at 6 h, for IL-8 at 24 h, and for TNF-α at 6 and 24 h did not show significant differences, as evidenced by *p*-values > 0.05. In addition, comparisons between all cytokine inductions at 6 and 24 h of stimulation by *C. glabrata* and by *C. albicans*, by *C. parapsilosis*, and by *C. tropicalis* also resulted in *p*-values > 0.05, indicating no significant differences.

Among the eight comparisons between all cytokines, the induction by resistant *C. albicans* and by resistant *C. parapsilosis*, four showed significant differences (*p*-values < 0.05). Specifically, three comparisons for IL-1β and TNF-α at 24 h favored *C. albicans*, while one comparison for IL-6 favored *C. parapsilosis*.

### 3.4. Comparison Between Cytokines Levels Induced by Different Susceptibility Phenotype of Candida Within Homogenous Species

Here, we evaluated cytokine levels according to fluconazole susceptibility phenotypes within individual *Candida* species, specifically *C. albicans* and *C. parapsilosis* (Table 2). For all four cytokines tested at both stimulation time points, *p*-values were <0.05, indicating significantly lower cytokine levels induced by susceptible strains compared with resistant strains. These findings confirmed the differences observed between the overall susceptible and resistant *Candida* strain groups.

### 3.5. Comparisons Between Cytokine Levels Induced by C. auris and Individual Candida Species, Fluconazole Susceptible and Resistant

Table 3 shows the forty-eight comparisons between cytokines levels after stimulation with *C. auris* and the different *Candida* species, both susceptible and resistant.

Among the thirty-two comparisons between the induction by *C. auris* and by susceptible strains, nearly 91% (twenty-nine) showed that cytokine levels induced by susceptible strains were significantly lower than by *C. auris*, with *p*-values < 0.05. However, IL-1β induction at 6 h by *C. auris* versus *C. albicans*, versus *C. parapsilosis*, and versus *C. glabrata* showed no significant differences (*p*-values > 0.05). All cytokine evaluations between both time points revealed significantly higher levels induced by *C. auris,* followed by *C. tropicalis*.

Among the sixteen comparisons between cytokines induced by *C. auris* and resistant strains, fourteen (87.5%) showed no significant differences (*p*-values > 0.05). The evaluation of TNF-α after 6 h stimulation by *C. auris* showed higher levels than by *C. parapsilosis* (*p*-value = 0.003), and after 24 h by *C. albicans* showed higher levels than by *C. auris* (*p*-value = 0.015).

### 3.6. Profiles of Cytokine Inductions According to Fluconazole Susceptibility Phenotype and Species of Stimulating Candida Strains

To further elucidate the biological significance of the dual effect observed in IL-1β production induced by resistant strains after 6 h of stimulation (Figure 2), we analyzed the responses elicited by individual *Candida* species among the resistant strains. Analysis of the data presented in Figure 4 shows that *C. parapsilosis* strains accounted for the observed variability in IL-1β induction. Further examination of this phenotype revealed a positive correlation between IL-1β levels at 6 h and the fluconazole MICs of these strains (Figure 5); strains with higher drug resistance induced higher IL-1β production (r = 0.674, *p*-value = 0.001). This IL-1β induction phenotype appeared to diminish after 24 h of stimulation (compare Figure 3 with Figure 6). Nonetheless, the correlation between IL-1β levels and fluconazole MICs remained statistically significant (Figure 7). No additional correlations between antifungal resistance and cytokine induction were observed.

Susceptible strains and *C. auris* induced lower amounts of IL-1β than TNF-α, with negative IL-1β/TNF-α ratios (0.52 from *C. albicans* and *C. parapsilosis*, 0.25 from *C. tropicalis*, 0.23 from *C. glabrata*, and 0.35 from *C. auris* stimulation). Resistant strains induced higher amounts of IL-1β than TNF-α, with positive IL-1β/TNF-α ratios (1.64 from *C. albicans* and 1.45 from *C. parapsilosis* stimulation).

After 24 hours of stimulation, inductions by susceptible strains and *C. auris* were still characterized by IL-1β levels lower than TNF-α, despite increasing or decreasing IL-1β/TNF-α ratios themselves (0.19 from *C. albicans*, 0.35 from *C. parapsilosis*, 0.21 from *C. tropicalis*, 0.28 from *C. glabrata*, and 0.56 from *C. auris* stimulation) (Figure 6). IL-1β/TNF-α ratios from resistant strain inductions changed to negative, similar to those obtained with the susceptible and *C. auris* strains, although the ratios were higher (0.90 from *C. albicans* and 0.80 from *C. parapsilosis* stimulation). It is apparent that the change was due to the remarkable increase in induction of TNF-α by resistant strains, not to the reduction in IL-1β quantity. Indeed, following prolonged stimulation, the quantified levels of IL-1β derived from *C. auris* strains remained significantly lower than those observed in the globally resistant strains (*p*-value < 0.05; specifically, *p*-value = 0.033). Conversely, the measured concentrations of TNF-α exhibited no statistically significant difference between the two groups under the same experimental conditions (*p*-value > 0.05; specifically, *p*-value = 0.615). These findings are visually represented in Figure 2.

## 4. Discussion

*Candida* colonizes the surfaces of healthy individuals as commensal but can cause life-threatening disseminated infections of endogenous origin in compromised and hospitalized hosts [4,5,27]. *C. albicans* can invade epithelial cell barriers, mostly of the gastrointestinal tract, through active endocytosis and a paracellular route involving the proteolytic digestion of tight junctions [3,4,28]. The early stages of *Candida*–host interactions correspond to the first defense response by the mucosal epithelial cells. These cells are equipped with PRRs and produce pro-inflammatory cytokines as effector weapons against the MAMP stimulus [7,29]. The fungal countermeasures towards host immune surveillance may be a critical point of transition from colonization to infection and are now considered part of the fitness attributes of yeasts [14]. Indeed, fungal fitness refers to the overall ability of a fungal organism to survive, grow, reproduce, and successfully interact with its environment, including the host, during infection. It encompasses a range of biological traits that contribute to the organism’s adaptability and competitiveness under various conditions and can be altered by the mechanisms underlying the antifungal resistance [21].

Our experiments, stimulating the human intestinal cell line DLD-1 with *Candida* strains differing in species and susceptibility phenotype to fluconazole, revealed preliminary but interesting results that we consider as possible starting points for further in-depth investigations.

The levels of cytokines induced by the fluconazole susceptible *Candida* were significantly different from and lower than those induced by *C. auris* strains, and the relation between the stimulating activity of these two groups was the same for all the tested cytokines, at the shorter and longer stimulation times. Differently, the relation between the stimulating activity of resistant strains (other than *C. auris*) and that of *C. auris* strains, and the relation between the stimulation activity of resistant and susceptible strains were different depending on the cytokine. IL-6 and IL-8 levels induced by the fluconazole susceptible *Candida* were significantly different from and lower than those induced by the resistant strains and by *C. auris*; IL-6 and IL-8 levels induced by the resistant strains and by *C. auris* were not different. The level of IL-1β obtained with the resistant strains was significantly different and higher than that obtained with *C. auris* at two stimulation times. The level of TNF-α obtained with the resistant strains was significantly lower than that obtained with *C. auris*, which was no longer evident after prolonged stimulation, giving the idea of delayed stimulation activity. The level of TNF-α obtained with the resistant strains was not different from that obtained with the sensitive strains at two stimulation times. These aspects were also represented by the profiles of induction and the relative ratios between diverse cytokines for each group of stimulating strains. The profiles obtained with susceptible strains and *C. auris* groups (see the most evident negative IL-1β/TNF-α ratio) were like each other over time, and different from those obtained with the group of resistant strains. In fact, resistant *Candida* strains gave profiles showing positive IL-1β/TNF-α ratios, being the level of the first cytokine higher than the second one.

The differences in immunostimulant ability that emerged within individual *Candida* species and among different susceptibility phenotypes suggest that the early host–parasite interaction can be considered a part of fungal fitness, which includes many aspects of their biology [29,30,31]. Vincent et al. first considered the fungus–host interaction as part of yeast fitness, having shown that resistance to amphotericin B in clinical and evolved-laboratory strains of *C. albicans* came at a high cost, i.e., reduced opposition to the host’s innate immune defense and less infectious damage than its susceptible counterpart [32]. In addition, to our best knowledge, this is the first study demonstrating this kind of relation concerning the very early host–pathogen immune response mediated by epithelial mucosal cells.

It is anyway hard to attribute the different behaviors of susceptible and resistant strains to advantages or disadvantages for the latter, given the fungal genetic plasticity that can translate mutational changes into gains of function [33,34,35]. In our study, the high level of cytokines induced by resistant strains could be beneficial to them by leading to high inflammation and permeability of mucosa lamina propria, in turn promoting microbial invasion into the favorable bloodstream [36,37,38]. Conversely, the ability of susceptible strains to modulate and inhibit mucosal defense signals could be impaired by mutations, responsible for resistance, which may come at a cost, as often observed among bacteria [39,40]. In our view, the different level and timing of IL-1β and TNF-α in comparison with IL-6 and IL-8 production by resistant strains, particularly by *C. parapsilosis*, might suggest that their stimulatory behavior is the result of a possible mutational suffering.

Another noteworthy point emerging from our data concerns *C. auris*. Assuming that fluconazole-susceptible *Candida* strains represent the wild-type phenotype, the similar stimulating profile induced by *C. auris* may also reflect a wild-type origin; its high fluconazole MICs, elevated cytokine induction levels, and genomic features reported in the literature may represent inherited traits of *C. auris.* Several studies have reported that, despite the presence of independently arising mutations, the distinction between intrinsic and acquired resistance among *C. auris* isolates is less clear than in other *Candida* species [13,41,42,43]. In this context, it is noteworthy that multidrug-resistant *C. auris* has emerged simultaneously on three continents [44].

Finally, our data permitted not to exclude a role for ergosterol in fungi–host interaction in animals, which has already been demonstrated in plants [9,10,11]. The cited work of Vincent et al. reinforces this possibility. They demonstrated that *Candida* strains resistant to amphotericin B, characterized by a reduced ergosterol content in the cell membrane, showed performance in parasite–host interactions and phagocytosis [12]. It can be envisaged that similar behaviors observed in this study could be due to other external fungal structures. A possible actor could be the wall glucan, already known to be a MAMP, whose folding might be improper, given the improper structure of the underlying fungal membrane rich in ergosterol. Structural differences between the glucans of fluconazole-susceptible and -resistant *Candida* strains may in turn result from independent nontarget effects induced by the stress response [45].

This study has limitations inherent to a preliminary investigation, but we hope it will serve as a foundation for more in-depth future research. Specifically, the absence of isogenic strains, resistance-related genomic data, and clade identification for *C. auris* [41] limits the accurate interpretation of the functional impact of resistance on fungal fitness at the strain level. Equally important and intended for future evaluation is the lack of phenotypic assessment of biochemical structures of the fungal membrane and cell wall. As previously discussed, mutations in genes involved in ergosterol biosynthesis can lead to drug resistance and overexpression due to gain-of-function, a possibility that must be evaluated [17].

Although this study requires further investigation, it offers valuable insights that enhance our understanding of fungal drug resistance. The diversity of fitness during the early immune response between fluconazole-sensitive and fluconazole-resistant *Candida* strains within the same species is evident. Another finding relates to the intriguing behavior of *C. auris* toward mammalian host cells in vitro; notably, this species appears to lie somewhere between fluconazole-sensitive and fluconazole-resistant *Candida* strains. Finally, our findings echo observations in plant cells on the role of ergosterol in interacting with host innate immunity.

## Figures and Tables

**Figure 1 microorganisms-13-01777-f001:**
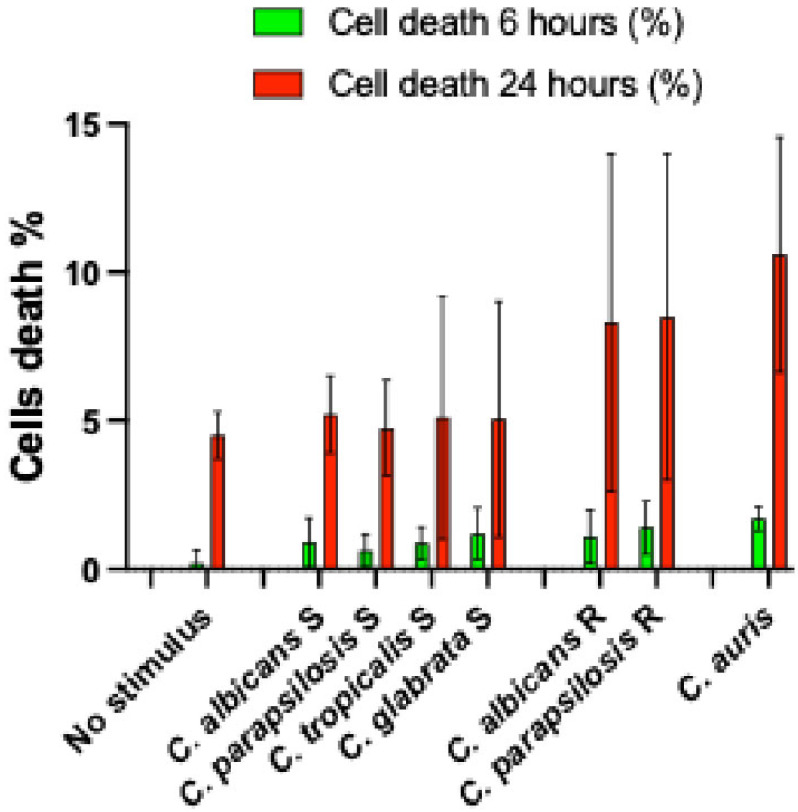
No significant differences in cell death were observed across experimental conditions. Sensitive (S) and resistant (R) fluconazole *Candida* strains. *P*—*p*-value (significant when <0.05); for all stimulatory condition, S 6 and 24 h, R 6 and 24 h, *p*-value > 0.05.

**Figure 2 microorganisms-13-01777-f002:**
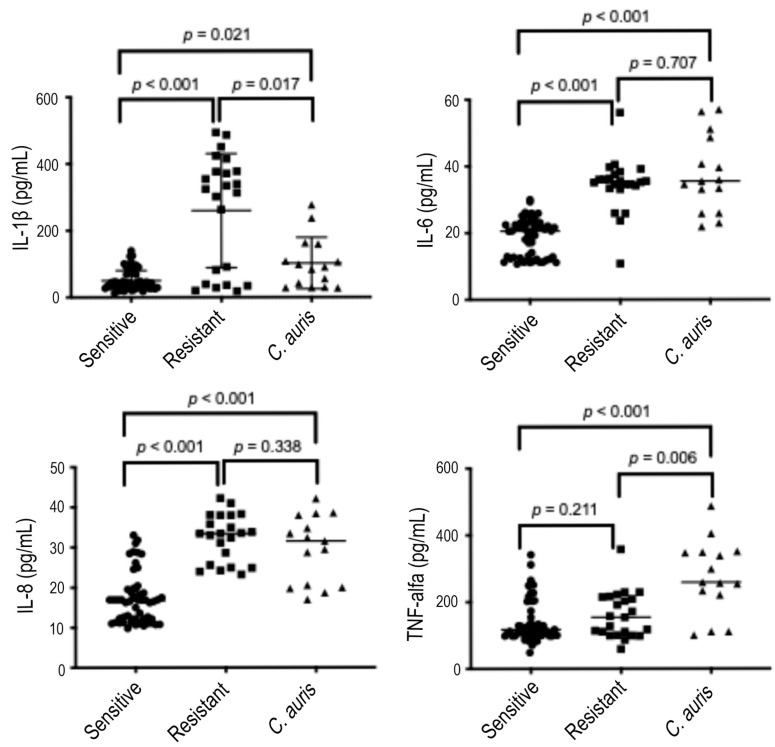
Cytokines produced after 6 h of stimulation. IL-1β—interleukin-1β; IL-6—interleukin-6; IL-8—interleukin-8; TNF-α—tumor necrosis factor α. Sensitive—fluconazole sensitive *Candida* strains; Resistant—fluconazole resistant *Candida* strains. *P*—*p*-value (significant when <0.05).

**Figure 3 microorganisms-13-01777-f003:**
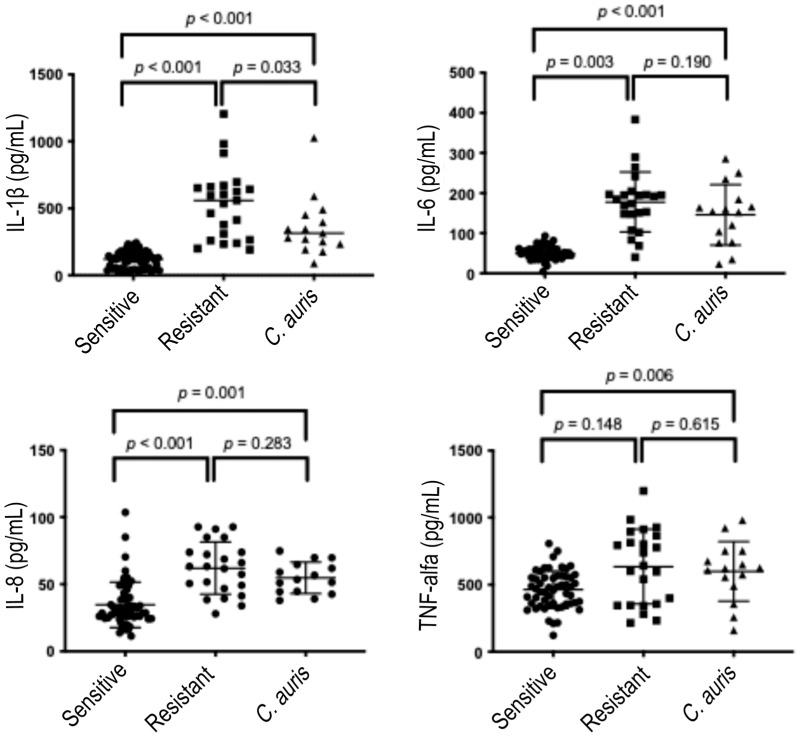
Cytokines produced after 24 h stimulation. IL-1β—interleukin-1β; IL-6—interleukin-6; IL-8—interleukin-8; TNF-α—tumor necrosis factor α; Sensitive— fluconazole sensitive *Candida* strains; Resistant—fluconazole resistant *Candida* strains. *P*—*p*-value (significant when <0.05).

**Figure 4 microorganisms-13-01777-f004:**
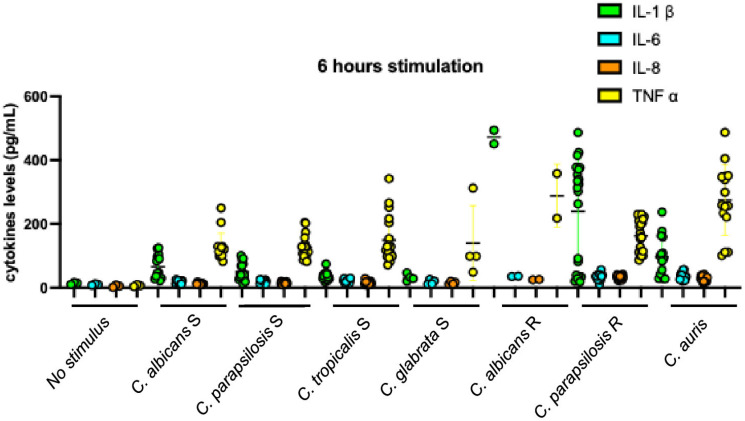
Cytokines produced after 6 h of stimulation according to species and fluconazole susceptibility phenotype of *Candida* stimulating strains. *C. albicans* S—*C. albicans* susceptible to fluconazole; *C. parapsilosis* S—*C. parapsilosis* susceptible to fluconazole; *C. tropicalis* S—*C. tropicalis* susceptible to fluconazole; *C. glabrata* S—*C. glabrata* susceptible to fluconazole; *C. albicans* R—*C. albicans* resistant to fluconazole; *C. parapsilosis* R—*C. parapsilosis* resistant to fluconazole IL-1β—interleukin-1β; IL-6—interleukin-6; IL-8—interleukin-8; TNF-α—tumor necrosis factor α.

**Figure 5 microorganisms-13-01777-f005:**
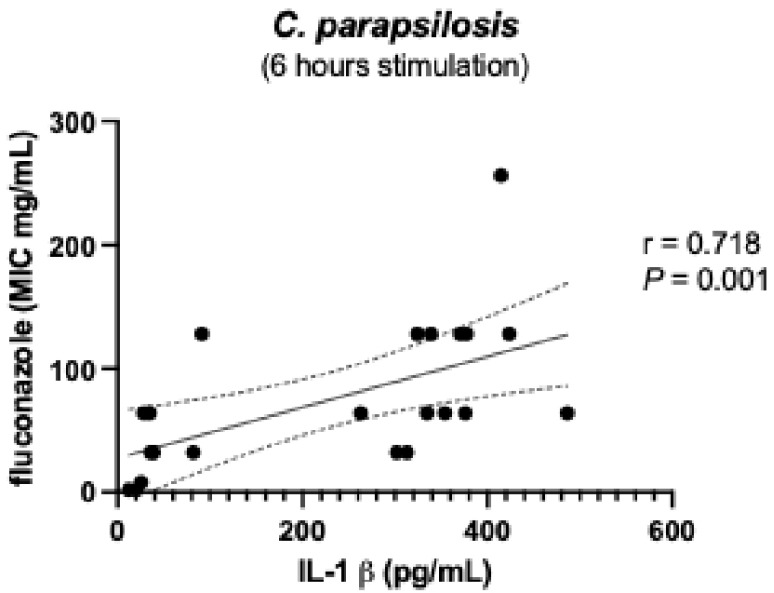
Positive correlation between IL-1 β levels induced after 6 h stimulation by *C. parapsilosis* resistant strains and their MIC to fluconazole. IL-1β—interleukin-1β. *p* < 0.05 significant. r—Spearman correlation; *P*—*p*-value (significant when <0.05).

**Figure 6 microorganisms-13-01777-f006:**
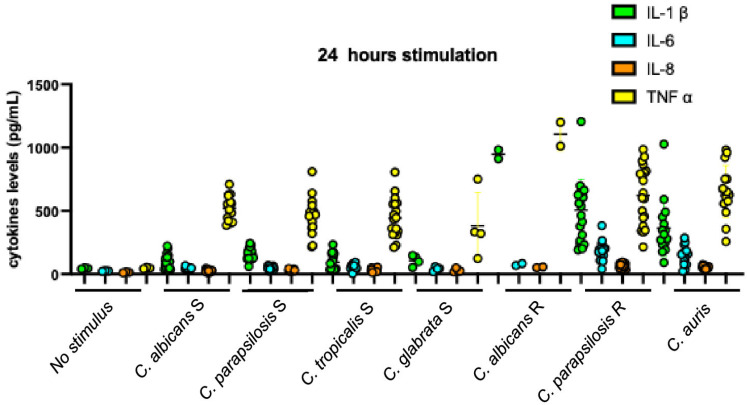
Cytokines produced after 24 h of stimulation according to species and fluconazole susceptibility phenotype of *Candida* stimulating strains. *C. albicans* S—*C. albicans* susceptible to fluconazole; *C. parapsilosis* S—*C. parapsilosis* susceptible to fluconazole; *C. tropicalis* S—*C. tropicalis* susceptible to fluconazole; *C. glabrata* S—*C. glabrata* susceptible to fluconazole; *C. albicans* R—*C. albicans* resistant to fluconazole; *C. parapsilosis* R—*C. parapsilosis* resistant to fluconazole; IL-1β—interleukin-1β; IL-6—interleukin-6; IL-8—interleukin-8; TNF-α—tumor necrosis factor α.

**Figure 7 microorganisms-13-01777-f007:**
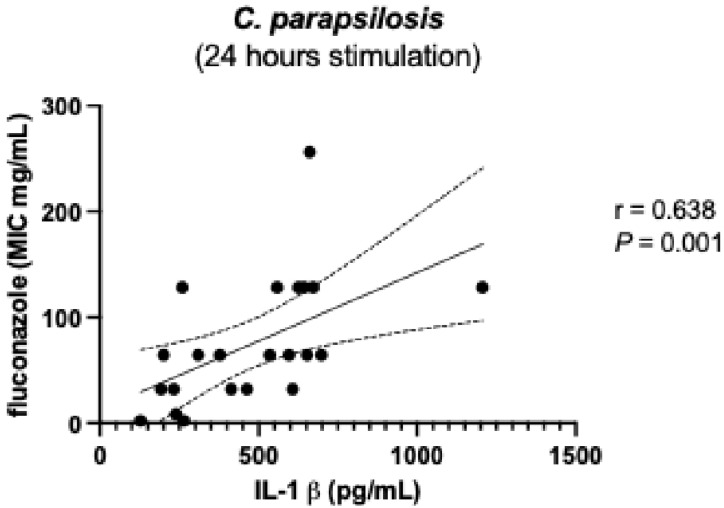
Positive correlation between IL-1 β levels induced after 24 h stimulation by *C. parapsilosis* resistant strains and their MIC to fluconazole. IL-1β—interleukin-1β. *p* < 0.05 significant. r—Spearman correlation; *P*—*p*-value (significant when <0.05).

**Table 1 microorganisms-13-01777-t001:** Susceptibility phenotypes of *Candida* strains.

*C. albicans*			*C. parapsilosis*						*C. tropicalis*			*C. glabrata*			*C. auris*				
N strain	MIC mg/L		N strain	MIC mg/L		N strain	MIC mg/L		N strain	MIC mg/L		N Strain	MIC mg/L		N strain	MIC mg/L			
	FZ	AB		FZ	AB		FZ	AB		FZ	AB		FZ	AB		FZ	AB	VOR	CAS
A1	1	1	P1	8	1	P21	32	1	T1	2	1	G1	2	0.25	Au1	>256	2	2	0.12
A2	0.5	1	P2	0.25	0.5	P22	32	2	T2	4	1	G2	2	1	Au2	>256	2	2	0.12
A3	2	0.5	P3	0.12	1	P23	32	2	T3	1	1	G3	0.5	0.25	Au3	>256	1	2	0.25
A4	0.25	0.5	P4	2	1	P24	2	4	T4	1	1	G4	2	0.5	Au4	>256	2	4	0.12
A5	0.25	1	P5	0.5	1	P25	64	0.5	T5	0.5	1				Au5	>256	2	2	0.12
A6	0.25	1	P6	0.5	0.25	P26	128	1	T6	0.5	1				Au6	>256	2	2	0.12
A7	0.5	0.5	P7	0.5	1	P27	64	0.5	T7	1	1				Au7	>256	2	2	0.12
A8	0.25	1	P8	0.5	1	P28	64	2	T8	1	0.5				Au8	>256	4	4	0.06
A9	0.25	1	P9	1	1	P29	128	2	T9	1	1				Au9	>256	2	2	>4
A10	0.5	0.5	P10	0.5	0.5	P30	128	1	T10	1	1				Au10	>256	2	4	0.12
A11	0.5	0.5	P11	0.5	0.5	P31	32	0.5	T11	0.5	0.5				Au11	>256	2	4	0.03
A12	0.5	1	P12	0.5	0.25	P32	32	0.5	T12	1	1				Au12	>256	2	4	0.25
A13	1	0.5	P13	0.5	0.5	P33	64	1	T13	2	1								
A14	<0.12	0.5	P14	0.5	0.5	P34	128	1	T14	1	2								
A15	0.25	1	P15	0.25	0.5	P35	256	0.5	T15	4	1								
A16	0.25	1	P16	0.5	0.5	P36	128	0.5	T16	2	1								
A17	>256	1	P17	1	1	P37	64	1	T17	1	1								
A18	>256	1	P18	1	1	P38	128	1	T18	1	1								
			P19	64	0.5	P39	64	1	T19	4	1								
			P20	16	0.5														

FZ—fluconazole; AB—amphotericin B; VOR—voriconazole; CAS—caspofungin.

**Table 2 microorganisms-13-01777-t002:** Comparison between cellular cytokine responses to stimulation with different *Candida* species.

	Stimulation By								
Cytokine and incubation time (hours)	Sensitive *C. albicans* vs.			Sensitive *C. parapsilosis* vs.		Sensitive *C. tropicalis* vs.	Sensitive *C. albicans* vs.	Sensitive *C. parapsilosis* vs.	Resistant *C. albicans*
	Sensitive *C. parapsilosis*	Sensitive *C. tropicalis*	Sensitive *C. glabrata*	Sensitive *C. tropicalis*	Sensitive *C. glabrata*	Sensitive *C. glabrata*	Resistant *C. albicans*	Resistant *C. parapsilosis*	Resistant *C. parapsilosis*
	*P*	*P*	*P*	*P*	*P*	*P*	*P*	*P*	*P*
IL-1β (6)	0.058	0.057	0.097	0.186	0.253	0456	0.001	0.013	0.012 *
IL-1β (24)	0.018 *	0.554	0.307	0.086	0.166	0.493	0.015	<0.001	0.032 *
IL-6 (6)	0.426	0.017 *	0.871	0.048 *	0.950	0.259	0.003	<0.001	0.707
IL-6 (24)	0.357	0.008 *	0.867	0.096	0.300	0.144	0.015	<0.001	0.032 *
IL-8 (6)	0.046 *	0.041 *	0.133	0.800	0.853	0.709	0.001	<0.001	0.216
IL-8 (24)	0.075	0.130	0.904	0.889	0.187	0.318	0.003	<0.001	0.557
TNF-α (6)	0.979	0.601	0.253	0.573	0.525	0.399	0.029	0.023	0.431
TNF-α (24)	0.929	0.054	0.152	0.090	0.195	0.403	0.041	0.043	0.008 *

IL-1 β (6)—interleukin-1β after 6 h stimulation; IL-1β (24)—interleukin-1β after 24 h stimulation; IL-6 (6)—interleukin-6 after 6 h stimulation; IL-6 (24)—interleukin-6 after 24 h stimulation; IL-8 (6)—interleukin-8 after 6 h stimulation; IL-8 (24)—interleukin-8 after 24 h stimulation; TNF-α (6)—tumor necrosis factor after 6 h stimulation; TNF-α (24)—tumor necrosis factor α after 24 h stimulation. Sensitive—fluconazole sensitive *Candida* strains; Resistant—fluconazole resistant *Candida* strains. *P*—*p*-value (significant when <0.05); * *p*-value < 0.05.

**Table 3 microorganisms-13-01777-t003:** Comparison between cellular cytokine responses to stimulation with *C. auris* strains and different *Candida* species.

Stimulation By						
Cytokine and incubation time (hours)	*C. auris* versus					
	Sensitive *C. albicans*	Sensitive *C. parapsilosis*	Sensitive *C. tropicalis*	Sensitive *C. glabrata*	Resistant *C. albicans*	Resistant *C. parapsilosis*
	*P*	*P*	*P*	*P*	*P*	*P*
IL-1β (6)	0.262 *	0.151 *	0.007	0.062 *	0.057	0.066
IL-1β (24)	<0.001	<0.001	<0.001	0.004	0.059	0.063
IL-6 (6)	<0.001	<0.001	0.004	0.002	0.941	0.663
IL-6 (24)	0.002	<0.001	0.005	0.014	0.235	0.088
IL-8 (6)	<0.001	<0.001	<0.001	0.040	0.529	0.230
IL-8 (24)	<0.001	<0.001	<0.001	0.049	0.941	0.235
TNF-α (6)	0.005	0.001	0.001	0.036	0.926	0.003 *
TNF-α (24)	0.030	0.002	0.001	0.077	0.015 *	0.981

IL-1β (6)—interleukin-1β after 6 h stimulation; IL-1β (24)—interleukin-1β after 24 h stimulation; IL-6 (6)—interleukin-6 after 6 h stimulation; IL-6 (24)—interleukin-6 after 24 h stimulation; IL-8 (6)—interleukin-8 after 6 h stimulation; IL-8 (24)—interleukin-8 after 24 h stimulation; TNF-α (6)—tumor necrosis factor after 6 h stimulation; TNF-α (24)—tumor necrosis factor α after 24 h stimulation. Sensitive—Candida strains sensitive to fluconazole; Resistant—Candida strains resistant to fluconazole. P—p-value (significant when <0.05); * p-value highlighted in the result section.

## Data Availability

The original contributions presented in this study are included in the article. Further inquiries can be directed to the corresponding authors.

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
