# Peer review of "The Role of Drug Resistance in Candida Inflammation and Fitness"

_microorganisms, 2025, doi:10.3390/microorganisms13081777_

Round 1

Reviewer 1 Report

Comments and Suggestions for Authors

This manuscript explores the relationship between fitness cost and antifungal resistance in various Candida species, a topic of high relevance to both clinical and microbiological research.

Major Comment:

I suggest that the authors include growth kinetics (e.g., in LB medium or a physiologically relevant medium like RPMI) as an additional indicator of fitness. This would strengthen the analysis by providing a more complete understanding of how resistance impacts growth under standard and host-simulating conditions.

Minor Suggestions:

    1. Please enhance the resolution of all figures in the manuscript to improve clarity and readability.

    2. IL-1β Data (24 h Time Point): In the Results section, consider stating that the two distinct populations observed at 6 hours in the resistant group appear to converge at 24 hours, with no visibly distinct populations.

    3. Tables 2 and 3 – Group Comparisons: Please clearly separate comparisons within the same category (e.g., resistant vs resistant) from those between categories (e.g., sensitive vs resistant) to avoid confusion.

    4. In Tables 2 and 3, review the p-values and asterisk (*) indicators. The manuscript notes significance at p < 0.005, but it's unclear if this is intentional or a typo for p < 0.05. Please revise and clarify.

    5. Section 3.1 – Resistance Cutoffs: Briefly mention the clinical breakpoints or cutoffs used to define resistance in Candida species. Specifically, clarify on what basis C. auris is categorized as resistant or sensitive, as this is currently unclear.

Author Response

Reviewer 1

Major Comment:

I suggest that the authors include growth kinetics (e.g., in LB medium or a physiologically relevant medium like RPMI) as an additional indicator of fitness. This would strengthen the analysis by providing a more complete understanding of how resistance impacts growth under standard and host-simulating conditions.

We thank the Reviewer for this suggestion. Accordingly, we have added the results in a figure.

Minor Suggestions:

  1.  
    1. Please enhance the resolution of all figures in the manuscript to improve clarity and readability.

We followed the suggestion by increasing the quality of the figures.

    1. IL-1β Data (24 h Time Point): In the Results section, consider stating that the two distinct populations observed at 6 hours in the resistant group appear to converge at 24 hours, with no visibly distinct populations.

In response to the suggestion, we have added a comment on this point in the manuscript.

    1. Tables 2 and 3 – Group Comparisons: Please clearly separate comparisons within the same category (e.g., resistant vs resistant) from those between categories (e.g., sensitive vs resistant) to avoid confusion.

In response to the suggestion, we reorganized the Table.

    1. In Tables 2 and 3, review the p-values and asterisk (*) indicators. The manuscript notes significance at p < 0.005, but it's unclear if this is intentional or a typo for p < 0.05. Please revise and clarify.

We thank the Reviewer for pointing out this error, which has now been corrected in the revised manuscript.

    1. Section 3.1 – Resistance Cutoffs: Briefly mention the clinical breakpoints or cutoffs used to define resistance in Candida species. Specifically, clarify on what basis C. auris is categorized as resistant or sensitive, as this is currently unclear.

In line with the reviewer’s suggestion, we have modified the text to better highlight and clarify the issue of Candida auris resistance and susceptibility.

Reviewer 2 Report

Comments and Suggestions for Authors

The aim of presented study was to compare the levels of cytokines produced by a colorectal adenocarcinoma cell line DLD-1 - CCL-221 stimulated with live yeast cells susceptible and resistant to fluconazole. The study was performed on an impressive number of 92 isolates and certainly required a lot of work and resources. The selection of some species, e.g. the inclusion of 4 strains of C. glabrata and 19 of C. tropicalis, which were 100% sensitive to the tested drugs is somewhat surprising. I suggest also supplementing the data regarding the origin of the strains, given that the study concerns intestinal cells, information on isolation from invasive infections or not, or from the gastrointestinal tract, may be relevant.

The ability of yeast to invade tissues and induce cytokine production varies between and within species. Therefore, in my opinion, the analysis should start by comparing the levels of cytokines produced after stimulation with different species, including C. auris (together susceptible and resistant strains of each species). Certainly, important differences will be discovered here. Subsequently, the  susceptibility phenotypes within the same species can be analysed (e.g. C. parapsilosis sensitive vs resistant) and finally the ability to stimulate cytokines by global fluconazole susceptibility phenotypes and different species combinations (e.g. C. auris vs fluconazole susceptible and resistant strains) may be compared.

2.3. Cell Culture - ...", as previously described" where? no reference given

Author Response

Reviewer 3

The aim of presented study was to compare the levels of cytokines produced by a colorectal adenocarcinoma cell line DLD-1 - CCL-221 stimulated with live yeast cells susceptible and resistant to fluconazole. The study was performed on an impressive number of 92 isolates and certainly required a lot of work and resources. The selection of some species, e.g. the inclusion of 4 strains of C. glabrata and 19 of C. tropicalis, which were 100% sensitive to the tested drugs is somewhat surprising. I suggest also supplementing the data regarding the origin of the strains, given that the study concerns intestinal cells, information on isolation from invasive infections or not, or from the gastrointestinal tract, may be relevant.

We thank the Reviewer for this insightful and encouraging comment. It serves as a strong motivation to continue our research in this field. Lastly, the suggested revision has been implemented in the manuscript, as recommended by the Reviewer.

The ability of yeast to invade tissues and induce cytokine production varies between and within species. Therefore, in my opinion, the analysis should start by comparing the levels of cytokines produced after stimulation with different species, including C. auris (together susceptible and resistant strains of each species). Certainly, important differences will be discovered here. Subsequently, the  susceptibility phenotypes within the same species can be analysed (e.g. C. parapsilosis sensitive vs resistant) and finally the ability to stimulate cytokines by global fluconazole susceptibility phenotypes and different species combinations (e.g. C. auris vs fluconazole susceptible and resistant strains) may be compared.

We thank the Reviewer for the thoughtful suggestion. However, we chose to structure the article as it is, because our study was specifically designed to investigate the relationship between fluconazole susceptibility phenotypes and the early host immune response. Our experimental approach grouped strains according to their susceptibility profiles (susceptible, resistant, and C. auris) rather than, or not only than, by species, in order to highlight the immunostimulatory differences associated with antifungal resistance. Moreover, C. auris was analyzed as a separate group due to its inherently high MICs and uncertain resistance classification, which complicates direct comparison with other species. This design allowed us to identify distinct cytokine induction patterns linked to resistance phenotypes, which may reflect differences in fungal fitness and host-pathogen interactions.

2.3. Cell Culture - ...", as previously described" where? no reference given

We have revised the text in accordance with this clarification request. Indeed, the explanation is provided later in the manuscript.

Round 2

Reviewer 2 Report

Comments and Suggestions for Authors

There are minor errors in the manuscript, e.g. lack of italics in Latin names, drug class names capitalized (96-98) and ambiguity (90 - “emocultures” - what does that mean?).

Author Response

There are minor errors in the manuscript, e.g. lack of italics in Latin names, drug class names capitalized (96-98) and ambiguity (90 - “emocultures” - what does that mean?).

We are grateful to the reviewer for highlighting these typographical errors. We have addressed and corrected them in the revised manuscript.